# Highly Diluted Glyphosate Mitigates Its Effects on *Artemia salina*: Physicochemical Implications

**DOI:** 10.3390/ijms24119478

**Published:** 2023-05-30

**Authors:** Mirian Yaeko Dias de Oliveira Nagai, Suham Nowrooz Mohammad, Andreia Adelaide G. Pinto, Ednar Nascimento Coimbra, Giovani Bravin Peres, Ivana Barbosa Suffredini, Maria Martha Bernardi, Alexander L. Tournier, Igor Jerman, Steven John Cartwright, Leoni Villano Bonamin

**Affiliations:** 1Research Center, Universidade Paulista, São Paulo 04026002, Brazilgiovani.peres@docente.unip.br (G.B.P.); ivana.suffredini@docente.unip.br (I.B.S.); maria.bernardi@docente.unip.br (M.M.B.); 2Faculty of Medicine, Universidade Federal de Alagoas, Maceió 57072900, Brazil; 3Institute of Complementary and Integrative Medicine, University of Bern, 3012 Bern, Switzerland; alexander.tournier@unibe.ch; 4BION Institute, 1000 Ljubljana, Slovenia; igor.jerman@bion.si; 5Cherwell Laboratory for Fundamental Research in Homeopathy, Oxford OX2 8NU, UK; steven.cartwright@oxford-homeopathy.org.uk

**Keywords:** brine shrimp, homeopathy, bio-resilience, hormesis, solvatochromic dyes, solvent polarity

## Abstract

Glyphosate is an herbicide widely used in agriculture but can present chronic toxicity in low concentrations. *Artemia salina* is a common bio-indicator of ecotoxicity; it was used herein as a model to evaluate the effect of highly diluted-succussed glyphosate (potentized glyphosate) in glyphosate-based herbicide (GBH) exposed living systems. *Artemia salina* cysts were kept in artificial seawater with 0.02% glyphosate (corresponding to 10% lethal concentration or LC10) under constant oxygenation, luminosity, and controlled temperature, to promote hatching in 48 h. Cysts were treated with 1% (*v*/*v*) potentized glyphosate in different dilution levels (Gly 6 cH, 30 cH, 200 cH) prepared the day before according to homeopathic techniques, using GBH from the same batch. Controls were unchallenged cysts, and cysts treated with succussed water or potentized vehicle. After 48 h, the number of born nauplii per 100 µL, nauplii vitality, and morphology were evaluated. The remaining seawater was used for physicochemical analyses using solvatochromic dyes. In a second set of experiments, Gly 6 cH treated cysts were observed under different degrees of salinity (50 to 100% seawater) and GBH concentrations (zero to LC 50); hatching and nauplii activity were recorded and analyzed using the ImageJ 1.52, plug-in Trackmate. The treatments were performed blind, and the codes were revealed after statistical analysis. Gly 6 cH increased nauplii vitality (*p* = 0.01) and improved the healthy/defective nauplii ratio (*p* = 0.005) but delayed hatching (*p* = 0.02). Overall, these results suggest Gly 6cH treatment promotes the emergence of the more GBH-resistant phenotype in the nauplii population. Also, Gly 6cH delays hatching, another useful survival mechanism in the presence of stress. Hatching arrest was most marked in 80% seawater when exposed to glyphosate at LC10. Water samples treated with Gly 6 cH showed specific interactions with solvatochromic dyes, mainly Coumarin 7, such that it appears to be a potential physicochemical marker for Gly 6 cH. In short, Gly 6 cH treatment appears to protect the *Artemia salina* population exposed to GBH at low concentrations.

## 1. Introduction

The industrial revolution was of paramount importance in humanity’s social and cultural development, but the environment began to be exploited. Advanced technologies often cause drastic changes in nature, with environmental pollution caused by industrial waste [1].

Knowing the environmental impact of technologies becomes imperative if prophylactic and corrective measures are to be taken. One of the ways to understand the toxic potential of substances is a set of tests with *Artemia salina* [2,3], also called brine shrimp. It is a micro brachiopod crustacean belonging to phylum Arthropoda, order Anostraca, which is present in different sizes and colors. Its diet is based on bacteria, unicellular algae, tiny protozoa, and organic debris, and the feeding is carried out by water filtration. The species is found in several regions of the planet, especially with a high degree of salinity and high temperatures. It is in the base of the trophic chain, being food for fish, and other crustaceans [4].

As part of its reproduction cycle, *Artemia salina* produces cysts in its habitat. Metabolically, the cysts are inactive while they are dry, but after being immersed in saline water, they hydrate and change from biconcave to spherical shape. Under favorable environmental conditions, the first layer of the cyst opens and exposes the embryo, which is surrounded by a membrane attached to the cyst (umbrella stage). At this stage, the nauplii complete their development and emerge as free swimmers [4]. *Artemia salina* is sensitive to several chemicals, such as phenolic compounds, nitrite, mercury [5], and organophosphate [6]. In a stressful environment, the cyst hatching is arrested, and the embryo remains in a quiescent state called diapause [7]. These features make *Artemia salina* a standard system for studying bio-resilience processes to environmental factors and, by extension, an excellent model for studying highly diluted (potentized) homeopathic preparations, whose main biological effects have been shown to facilitate resilience processes in different living systems [8,9].

Glyphosate [*N*-(phosphonomethyl) glycine] is a non-specific organophosphate that inhibits the enzyme 5-enolpyruvylshikimate-3-phosphate synthase, promoting the interruption of the essential aromatic amino acid syntheses in plants [10]. It is one of the most common herbicides used worldwide, traces of which can be found widespread in many different species [11,12,13], raising concerns about its possible action as an endocrine disruptor when present in commercial preparations at low concentrations [13,14,15].

The environmental impact of herbicides, in general, and glyphosate, in particular, has been a concern. Glyphosate can persist in seawater under low-light conditions, which implies a potential persistence of this molecule in the sediment [12,13], probably affecting the aquatic fauna and other terrestrial species. Recently, glyphosate was associated with amphibian and mammal metabolic disturbances, such as delay in *Xenopus laevis* oocyte maturation [14] and lipid metabolism disruption in the offspring of mice due to changes in critical gene expression [15].

Such concerns about glyphosate also apply to other herbicides and pesticides (mixtures of one or more active substances formulated to protect plants from harmful organisms, also called phytosanitary products) commonly used in agriculture. The harmful potential of pesticides for the environment and human health is recognized by international regulatory agencies. In this way, the “Sustainable Use of Pesticides Directive” (SUD) was adopted in 2009 as one of the follow-up actions of the European Commission’s thematic strategy on the sustainable use of pesticides to establish goals for the implementation of systems fairer, healthier, and more ecological food production, reducing risks and impacts. An updated proposal for the regulation of the use of phytosanitary products in the European Community was proposed in 2022 [16].

The use of potentized high dilutions (serial highly diluted substances submitted to vigorous agitation) in aquatic animals has been reported in clinical and experimental studies over the last 20 years; a recent review of the field was published recently [17]. However, the use of potentized preparations in assisting living systems in recovering from exposure to chemical agents is still a matter of speculation. The literature on this topic is restricted to clinical use [18,19]. Still, its environmental impact is poorly studied except for recent studies which involved the use of homeopathic products in natural water sources for agricultural purposes [20,21,22]. The tracking of Phosphorus 30 cH activity in a set of lakes was possible due to the spectrophotometry analysis of water samples using solvatochromic dyes [23].

This approach opens new possibilities for the rational use of potentized high dilutions as a valuable tool for the mitigation of the effects of environmental toxins. In addition, results from the use of solvatochromic dyes may provide support for the quantum electrodynamic (QED) based hypothesis of water electric dipole aggregates being a key element for the mechanisms of action of potentized high dilutions [24].

### Objective

Under laboratory conditions, this study aimed to investigate the putative mitigating activity of highly diluted potencies of glyphosate-based herbicide (GBH) on *Artemia salina* cysts exposed to low concentrations of GBH. Nauplii development and physicochemical parameters of water were analyzed.

## 2. Results

### 2.1. Analysis of the Nauplii Behavior and Vitality

Nauplii exposed to GBH LC10 showed lower vitality at 48 and 72 h, expressed by the proportion of nauplii that swam at the bottom compared to the surface of the water column. This effect was seen in the groups treated with succussed water, Gly 30 cH, and Gly 200 cH. Nauplii treated with *Ethilicum* 1 cH presented some protective effect, expressing a behavior similar to that of unchallenged nauplii. However, when nauplii were treated with Gly 6 cH, they experienced significant protection at 48 h. However, this effect was transient since it was no longer evident after 72 h (Figure 1).

### 2.2. Analysis of Nauplii Morphology

The healthy/defective nauplii ratio was higher in the groups treated with Gly 6 cH (*p* = 0.036) and Gly 30 cH (*p* < 0.0001) in relation to the unchallenged group, and challenged groups treated with succussed water or *Ethilicum* 1 cH (controls). It is also noted that the simple exposure of nauplii to GBH LC10 did not produce teratogenic effects, according to the results presented by the groups treated with succussed water and the unchallenged group, with no statistical significance between them (Figure 2).

### 2.3. Analysis of the Hatching Rate

The relative population of live nauplii showed that the treatment with Gly 6 CH and Gly 200 CH significantly reduced the hatching rate in relation to the unchallenged group after 48 h of cyst hatching (*p* ≤ 0.02). Data are expressed in Figure 3.

### 2.4. Analysis of Hatching According to the Treatment and Salinity

The relative population of each nauplii stage after 48 h for different levels of salinity and GBH exposure conditions are represented in Figure 4, Figure 5, Figure 6 and Figure 7.

Figure 4 shows the percentage of nauplii cysts at the “umbrella” stage and living active nauplii after 48 h of hatching under different experimental conditions, whatever the GBH concentration. As expected, there was no significant difference in cyst percentage among the treatment conditions since the cyst suspension was homogenized before seeding. However, there was a significant increase (*p* < 0.05) in the rate of “umbrella” stage nauplii after the treatment with Gly 6 cH in 80% seawater salinity in relation to the other conditions (B). A mild and non-significant decrease in the percentage of living active nauplii was also seen after the treatment with Gly 6 cH in an environment corresponding to 90%, 80%, or 60% seawater salinity.

### 2.5. Analysis of Hatching According to the Intensity of the Challenge

The relative population of each stage after 48 h of the hatching process was evaluated according to the glyphosate (active principle) concentration, that is, zero, LC10 (0.02%), LC30 (0.03%), and LC50 (0.05%), whatever the treatment (Figure 5). The challenge with GBH LC10 stood out compared to the other concentrations, as there was a lower percentage of unhatched cysts (A) and a higher rate of living active nauplii compared to the different challenge levels (C), being both complementary data. No difference among the levels of glyphosate challenges was seen regarding the percentage of the “umbrella” stage (B). The results show a non-linear behavior of GBH-inducing hatching, making this effect more effective at LC10. There was no concentration-dependent response in this case.

### 2.6. Analysis of Nauplii Activity

Figure 6 shows the frequency of locomotion and stops of nauplii as a measure of general activity in respect of the level of water salinity in samples treated with Gly 6 cH (A and B) and GBH exposure (C and D).

No difference among groups was seen in relation to water salinity and Gly 6 cH treatment (A and B). However, the exposition of nauplii to GBH LC50 (glyphosate 0.05%) produced a trend to increase in locomotion (C) and a significant increase in the stopping frequency (D). There was no statistical interaction between salinity/treatment and GBH exposition levels in both variables: locomotion [F _(18, 210)_ = 0.690, *p* = 0.818] and stopping [F _(18, 210)_ = 1.090, *p* = 0.364].

### 2.7. Physicochemical Analysis with Solvatochromic Dyes

#### 2.7.1. Experiment 1

The solvatochromic dyes that interacted with seawater from cysts treated with Gly 6 cH were Coumarin 7 and ET33. Methylene violet interacted with seawater from cysts treated with Gly 200 cH only (Figure 7). The other dyes tested (Nile red, Rhodanine, ET30, NN-DMIA) showed no interaction with seawater from cysts treated with any potencies of GBH.

Solvatochromic dyes respond to homeopathic potencies through an increase in their electronic polarization. Changes in their spectra reflect this increase in polarization, but absorbances can increase or decrease according to a dye’s particular electronic structure and aggregation characteristics in solution.

#### 2.7.2. Experiment 2

The dyes Coumarin 7 and ET 33 were the most responsive to samples treated with Gly 6 cH from experiment 1. Hence these dyes were selected to analyze the water samples obtained from experiment 2, chosen from the most significative results related to different salinity levels and glyphosate concentrations (unexposed, CL 10 and CL50) (Figure 8).

As seen in Figure 8, the absorbance of Coumarin was highest with water samples of 60% and 80% seawater salinity from cysts treated with Gly 6 cH that had not been exposed to GBH (A). Under the same conditions but with cysts that had been exposed to 0.02% glyphosate (LC10), the absorbance of Coumarin 7 was highest, with levels of salinity of 60%, 80%, and 90% salinity (B). With cysts that had been exposed to 0.05% glyphosate (LC50) and treated with Gly 6 cH, the highest absorbances were seen with 80% and 90% salinity, with lower but still raised absorbances at 50% and 60% salinity (C). Seawater that had not been involved in any manipulations provided the main reference (baseline) in all cases.

The absorbance of ET33 under different experimental conditions of GBH exposure was higher only in water of 90% salinity from cysts exposed to 0.05% glyphosate (LC50) and treated with Gly 6 cH (Figure 9C).

The data from the solvatochromic dye experiments point to Coumarin 7 as an excellent physicochemical marker for Gly 6 cH, even under different conditions of salinity and glyphosate concentration. It seems that the higher the glyphosate concentration, the more responsive Coumarin 7 is to Gly 6 cH, whatever the salt concentration. ET33 showed a narrower response profile, highly dependent on glyphosate concentration.

## 3. Discussion

The results show protective effects of Gly 6 cH after hatching, with an increase in vitality and an improved healthy/defective nauplii ratio. The hatching rate was also lower, with retention of nauplii development in the umbrella stage. Such a reduction in the hatching rate suggests this is a survival mechanism of nauplii in non-ideal environments. The result in this study mirrors the results previously described in nauplii exposed to mercuric chloride LC10 and treated with homeopathic potencies of mercuric chloride in a similar experimental model [8]. The effects seen in both studies involving nauplii reflect an adaptive characteristic of the species, in which cysts can be considered resistant forms [7,25,26]. Therefore, a lower hatching rate could be seen as a bio-resilience marker for homeopathic potencies.

The salinity level seems significant concerning the degree of effectiveness of Gly 6 cH since the biological effects were more evident at 80% seawater salinity. It was also seen in the interaction between water samples and solvatochromic dyes, with a maximum concentration-dependent effect at 80% salinity. The presence of ionic species in solution has been observed to affect NMR relaxation values associated with homeopathic potencies in a previous study [27]. As yet, no satisfactory explanation for this phenomenon has been established other than a suggested mechanism [28]. The current research shows that the relationship between potency action and ionic strength is not simple, given that Gly 6 cH effectiveness also depends on glyphosate concentrations, as shown in Figure 8. The effects of Gly 6 cH on cyst hatching were more evident when cysts were exposed to low concentrations of glyphosate (LC10), a situation that mimics chronic environmental exposure in rural areas, where GBH is routinely used in agriculture [29,30,31].

The toxicity of GBH was more evident under the LC50 challenge, with an increase of nauplii locomotion regardless of the treatment or degree of salinity. Nevertheless, subtle but biologically significant effects were observed in the tests in which LC10 was used. No teratogenic effects of GBH were observed when cysts were exposed to LC10, although the selective effects of Gly 6 cH and Gly 30 cH were quite evident in this condition. These findings show an interesting potential for Gly 6 cH as an additional layer to reduce possible ecological disruptions produced by this class of agrochemicals. Thus, future real-life field studies should be performed to evaluate its usefulness in large-scale agriculture conditions, as seen in other experimental models [22].

The way of action of Gly 6 cH on nauplii development seems to be related to mechanisms of environmental adaptation, such as the gene expression of bio-resilience markers, as described in other aquatic species [32]. Nevertheless, the final glyphosate concentration in Gly 6 cH is too low (5.9 × 10^−13^ M) to produce classical biochemical effects when diluted 1:10 and 1:100 in water (reaching about 6 × 10^−14^ M and 6 × 10^−15^ M), indicating the necessity to consider alternative hypotheses about its possible mechanism of action. In parallel, the influence of the salinity on the Gly 6 cH effects and the corresponding interaction between treated water and a specific solvatochromic dye (Coumarin 7) are two points suggesting that the physicochemical properties of highly diluted substances might be relevant aspects.

Whilst the mechanism of action of potentized high dilutions is still unknown, several hypotheses have been proposed in the last decade. These include hypotheses based on pharmacologically active nanoparticles, nanobubbles, the exclusion zone of water, and so-called “nano associates” [27,33,34,35].

In particular, the hypothesis of coherence domains proposed by Del Giudice and collaborators in 1988 [36] could be considered one of the most plausible explanations. Following this hypothesis, it is assumed there are specifically ordered molecular aggregates that make up the mesoscopic phase of water [37,38,39]. This hypothesis is based on a quantum electrodynamic (QED) model, which predicts that quantum electrodynamic effects could lead to the formation of local resonance structures in water [36], which would be reinforced by the presence of ions [37]. This QED model could serve as a theoretical background for understanding the imprinting of molecular information from a starting solution (such as a glyphosate solution) in a polar liquid medium such as water via coherent domains of different types [40,41].

Hence the dipole moment of water might be implicated in the formation of various types of coherent domains, depending on the solute present [24]. The formation of these structures would lock in electromagnetic frequencies coming from its environment, thereby providing a sort of fingerprint (specific memory) of the original substance [42,43,44,45,46,47]. The prevalence of a specific coherent domain pattern could be seen as a fingerprint of Gly 6 cH, for instance, and its presence, therefore, could be indirectly indicated by its specific interaction with the solvatochromic dye Coumarin 7.

Some experimental evidence supports this theory. For example, solvatochromic dyes have been shown to act as probes that track the presence of homeopathic potencies [23,46]. Assays using solvatochromic dyes are well established, and several publications are available describing the evolution of the technique [47,48,49,50,51]. Solvatochromic dyes become more polarized in the presence of homeopathic potencies, and this appears to be a dye and potency-specific phenomenon, with some dyes responding to a particular potency and others not. Potencies may therefore have particular molecular fingerprints. The model described above is consistent with the biological particularities of high dilution effects. However, more work needs to be undertaken to confirm this hypothesis.

The use of high dilutions to mitigate chronic or acute intoxication is not new as a therapeutic resource. References to this experimental approach are available in the literature [52,53,54,55,56]. However, a recent focus on its potential use in agriculture [56] and aquatic ecosystem management [8,17] requires further studies. It is known that the concentration of ions in aquatic ecosystems has been changing worldwide. Salt inputs of anthropogenic origin are salinizing a large part of freshwater ecosystems. Song and Brown, 2006, provided insight into how the susceptibility of a species to an insecticide can be affected by changes in salinity concentrations [57]. According to Evans and Kültz, 2020, salinity stress is linked to the abrupt change in salt concentration in the environment, which can occur due to different factors such as tidal flow, storms, droughts, or evaporation from small bodies of water. However, gradual changes in saline concentration can also cause osmotic stress in aquatic habitats if levels exceed limits that decrease the survival capacity of resident organisms [58].

Although the exact mechanism of action of homeopathic high dilutions is still unknown, it may involve electromagnetic fields or polar/charged imprints in water (or other polar solvents). This possibility, along with a previous study that seemed to show that dissolved salts affect the NMR relaxation values associated with homeopathic potencies [27], prompted an examination of the effect of varying the level of salinity (ionic strength) in solutions in which *Artemia salina* was challenged with GBH along with potencies thereof. The fact that mitigation of the effects of GBH by potentized highly diluted GBH was dependent on the level of salinity in solution may be important, not only in furthering our understanding of the mechanism of action of homeopathic high dilutions but also in designing future field studies in aquatic environments.

## 4. Materials and Methods

### 4.1. Ethics

This research was not eligible to be submitted to CEUA (Committee on Ethics in the Use of Animals) approval since only non-sentient invertebrate animals were used, following the CONCEA (National Council for the Control of Animal Experiments—Brazil) and EU Directive 2010/63/EU for animal experiments standards.

*Artemia salina* is a microcrustacean commonly used as an alternative method to replace rodents and other sentient species for experimental studies. The nervous system in this species is very primitive, composed of two nervous ganglion chains linked to one another by axons. Only vital functions and movements are controlled; thus, the reaction to environmental stimuli is only a reflex, with no emotionality since they have no limbic system or similar structure responsible for it. Similarly, nervous ganglions participate only in autonomic functions in vertebrates [59].

In the present study, the protective effects of highly diluted potencies of GBH (Gly) on *Artemia salina* were investigated to test whether it would mitigate the impact of low water concentrations of this organophosphate on embryo and larva development, mimicking real-life environmental conditions. The model allows extrapolation of the results to marine and other salt-free biomes, including other species and humans.

### 4.2. Potentized Glyphosate Preparation

According to the Brazilian Homeopathic Pharmacopoeia, 3rd Edition [60], homeopathic preparations of GBH must be classified as a hetero-iso-therapeutic, whose active principles are external to the patient, along with allergens, food, cosmetics, medicines, toxins, dust, pollen, and solvents, amongst others. The potentiation process refers to serial 1:100 dilutions followed by 100 vigorous vertical shaking of the liquid. Thus, the nomenclature cH refers to the number of centesimal dilutions and shaking cycles.

A GBH was purchased as a commercial formula (10% aqueous solution or 0.59 M) named GLIFOMATO^®^ (Insetimax, Jardinópolis, Brazil). Glyphosate (Gly) 6 cH, Gly 30 cH, Gly 200 cH, that is potentized GBH in different glyphosate dilution levels (which theoretical concentrations are 5.9 × 10^−13^ M, 5.9 × 10^−61^ M, 5.9 × 10^−401^ M respectively), and *Ethilicum* 1 cH (vehicle 0.15 M or potentized 30% hydro-alcoholic solution used as a comparative control) were prepared the day before using sterile purified Type 1 water (18.2 MΩ•cm at 25 °C) obtained from a Direct-Q3 purification system (SmartPark Direct Q3) with Biopak filters (Millipore, Darmstadt, Germany), and manipulated in a laminar flow cabinet. The working potencies (6 cH, 30 cH, and 200 cH) were prepared in 30% hydro-alcoholic solution from stock lower potencies (5 cH, 29 cH, and 199 cH) obtained from a magistral homeopathic pharmacy registered in the Brazilian National Agency for Sanitary Vigilance—ANVISA.

Sterile conventional amber type 2 glass vials were used, in which 100 µL of each stock potency was poured into 9.9 mL of pure, sterile water. All potentiation procedures were carried out under a laminar flow cabin, and the final dilution was vertically succussed 100 times using a mechanical arm (DENISE-AUTIC, São Paulo, Brazil) before being used. One flask containing 10 mL of pure, sterile water was also submitted to succussion in the same device and used as a negative control.

### 4.3. Experimental Design

#### 4.3.1. Experiment 1

Reconstituted seawater was prepared by diluting 30 g of sea salt (Ocean Fish—PRODAC, Citadella PD, Italy) in 1000 mL of sterile distilled water under constant agitation. In large culture bottles, 0.6 g of *Artemia salina* cysts (Global Biotérios, São Paulo, Brazil), kept dry up to be used (290,000 units per gram), were added to 396 mL of artificial seawater containing a suspension of 0.06% *Saccharomyces cerevisiae* as food for the nauplii after the depletion of the yolk reserves. Standard procedures of oxygenation and constant luminosity were provided to promote the cysts hatching in 48 h (Figure 10). Based on previous studies [61], a plate of mu-metal was used to separate one bottle from another to avoid eventual field-based communication among them without changing the light incidence on the bottles. The bottles were not isolated from the environment, the magnetic field was very low and constant in different positions on the shelf, independent of the mu-metal plates. The room features favored magnetic field isolation, whose measurements made at 50 Hz (Smart-Sensor AS 1392, Intel Instruments, Singapore) showed an average of 0.06–0.07 µT in different parts. Temperature and room humidity were verified daily.

Challenging cultures with GBH was carried out simultaneously with cyst insertion into the water to give a final glyphosate concentration of 0.02% (*v*/*v*), equivalent to the LC10 (lethal concentration for 10% of exposed 48 h-born nauplii). This concentration was chosen to mimic situations of chronic environmental exposure, especially in rural areas where GBH is used [11,12,62,63]. Different potencies of GBH were simultaneously added to give a concentration of 1% of the total volume of water (or 4 mL). Six treatments were set at this stage: 1—unchallenged (cysts neither exposed to GBH nor treated with potencies of GBH); 2—exposed to 0.02% glyphosate (active principle) and treated with succussed water; 3—exposed to 0.02% glyphosate and treated with *Ethilicum* 1 cH; 4—exposed to 0.02% glyphosate and treated with Gly 6 cH; 5—exposed to 0.02% glyphosate and treated with Gly 30 cH; 6—exposed to 0.02% glyphosate and treated with Gly 200 cH. Treatments were performed blindly and just once. The codes were revealed after statistical analysis.

At the end of 48 h, the liquid containing the initial stages (I to V) of nauplii hatching was collected in a Beaker. The content was gently and constantly homogenized, and an aliquot of 100 mL was filtered using filter paper to separate the biological samples and the remaining water for further physicochemical analysis. The unfiltered portion of the living nauplii was distributed in three equal parts to be used in different experimental assays.

In the first assay, the first portion of the water containing living nauplii was transferred to 50 mL Falcon tubes with screw caps for euthanasia and kept overnight under anoxia conditions. The following day, the nauplii deposited at the bottom of the tube were carefully collected with surrounding water using a wide-tip plastic Pasteur pipette and immediately placed on clean histological slides to avoid damaging the specimens by handling. The material deposited on each slide was gently covered with a clean coverslip and the remaining drop of water, then sealed with enamel to preserve the space between the coverslip and the slide without compressing the larvae. Three slides were prepared for each experimental group to evaluate the morphological features. The microscopic fields covering the whole coverslip area were registered at 4× magnification within 24 h. The digital photographs were analyzed by the average number of nauplii per field and the number of healthy and malformation-bearing nauplii. The total number of nauplii analyzed in this assay was 1334. The most common malformations observed were: 1—normal; 2—lack of one antennule; 3—lack of both antennules; 4—lack of antenna (Figure 11).

In the second assay, the second portion was used for behavioral analysis. Nauplii were transferred with a Pasteur pipette from the Beaker under constant agitation to acrylic transparent tubes (BD, Franklin Lakes, NJ, USA) containing 5 mL of seawater free of *Saccharomyces cerevisiae.* Ten nauplii were transferred per tube, and six tubes were used per group, three were evaluated after 38 h, and three were evaluated after 72 h. The distribution of living nauplii in the water column was used as a vitality parameter. Thus, the nauplii swimming next to the surface are more active and healthier, searching for oxygen. Nauplii swimming next to the tube bottom are less active, or immobile, which is considered a signal of death. Thus, the water column was divided into five equal parts, each corresponding to 5 mL. The scale was marked on the outer face of each tube. The surface column was defined as the 3/5th of the column closest to the water surface, and the bottom column was defined as the 2/5th part of the column at the bottom of the tube, which included dead and low vitality swimming nauplii. Then, the number of nauplii swimming in each column (surface or base) was recorded in the function of time (48 and 72 h time points). The total observed in this assay was 403. Dead nauplii were not counted.

In a third assay, the third portion was transferred to a 96-well plate, 100 µL per well, under constant homogenization. Such samples were used to calculate the relative population of nauplii born per treatment (live nauplii/100 µL) after 48 and 72 h of hatching. The viability criterion was naupliar mobility. Samples from each treatment were distributed in 12 wells, a total of 59 nauplii.

#### 4.3.2. Experiment 2

The second analysis phase was performed in a series of 96-well plates (8 rows of 12 wells each) on consecutive days. Five to eight cysts were distributed per well in 100 µL of saline water containing food (0.06% *Saccharomyces cerevisiae*). To obtain this population density, 75 mg of cysts were suspended in 200 mL of seawater with food, kept in constant agitation to be homogenized, and immediately distributed into the wells.

The potency chosen for this phase was Gly 6 cH, given the results obtained in experiment 1. The objective was to identify variables affecting the action of Gly 6 cH with respect to the hatching process and newborn nauplii behavior. The variables investigated were the level of water salinity and the concentration of glyphosate used to challenge cysts using standard LCs (lethal concentrations). It should be noted that the LCs were established by exposing 48 h-born nauplii, counting dead/live units after 24 h, and discounting the percentage of death in the baseline group.

Two independent sets of 4 plates were prepared. Seven rows of 12 wells were used per plate. Each row contained 200 µL of seawater per well at different salinity levels with food and nauplii, with or without Gly 6 cH. The experimental design is shown in Table 1.

Due to the potential sensitivity of potentized highly diluted substances to electromagnetic fields [24], all plate assays were performed in a hand-made Faraday cage illuminated 24 h a day with fluorescent light to allow maximum cysts to hatch. The exposure to light was homogeneously distributed among the plates, being the lamp 30 centimeter from the plates´ surface. The EMF near the shelves varied from 0.01 to 0.2 μT at 50 Hz (Smart-Sensor AS 1392, Intel Instruments, Singapore). The plates remained open to enable complete oxygenation of the wells. Two series of plates were prepared on consecutive days, and analyses were performed after 48 h. A total of 48,749 living nauplii were analyzed in this assay.

The nauplii stage and locomotion were recorded in a 10-s video for each well. The analysis of the videos allowed the counting of cysts, nauplii in the “umbrella” stage (still coupled to the cyst wall), and live swimming nauplii (from instar I to instar V, taken together). The ratio (percentage) between each main stage (cysts, umbrella, nauplii) and the total number of individuals recorded per well were measured for statistical analysis. Since the nauplii structural development was not the focus of this essay, but just their mobility, the differentiation of instar I to V was not performed. The number of stops and the movement (or tracking) of each nauplius were also identified, being the locomotion measured in pixels. Video recordings were made in AVI format using ImageJ 1.52 software, and the Trackmate Image J Plug-in automatically calculated the mobility in each well. In this case, the adjustment for reading RGB colors was set, standardizing the yellowish color of the nauplii as a reference for tracking mobility. The images were transformed into 8-bit binary resolution and analyzed by the Phansalkar method [64] included in the Image J Plug-in. The data presented in the Track Statistics worksheet were finally copied to a Microsoft Excel sheet. The Track Stop average (number of nauplii stops) and the Track Displacement average (nauplii movement) were calculated.

### 4.4. Physicochemical Analysis of Water

#### 4.4.1. Sample Preparation for Analysis and Coding

Water samples obtained from experiments 1 and 2 were frozen at −20 °C and thawed on the eve of physicochemical analysis. These samples were prepared for physicochemical as follows: 30 mL amber glass vials (previously autoclaved) were filled with 9.9 mL of pure, sterile water under laminar flow. Then, 100 µL of each sample was added to each bottle. The bottles were closed, identified, and submitted to 100 automatic succussions in a mechanical arm (Denise-AUTIC, São Paulo, Brazil). Two flasks were filled with 10 mL of purified water without any dilution for control, one submitted to succussions, and the other was left at rest.

Before the analysis, all dilutions were filtered through a 0.22-micrometer mesh filter (MILLIPORE, Darmstadt, Germany), passing the contents into a second sterile vial, always in the laminar flow cabin. The second vial was labeled and coded by a person from the laboratory’s technical team not involved in any stage of the study. So, all the physicochemical analyses were also performed blind.

#### 4.4.2. Spectrophotometry with Solvatochromic Dyes

The dyes used were prepared the day before use by diluting them in absolute ethanol [Synth, Diadema, Brazil] at concentrations previously stipulated [48,49,50] and let to rest for stabilization (Table 2). The entire contents were transferred to sterile Falcon tubes tightly covered in aluminum foil to protect them from light. The following day, in a laminar flow cabin, each dye was filtered in a 0.22-micrometer mesh filter (MILLIPORE, Darmstadt, Germany) just before use. For sample analysis, 25 µL of each control or potency solution from experiments 1 and 2, as prepared and described above, was added to each cuvette containing 1475 µL of dye solution so that the final sample/dye dilution had a 1:60 ratio. The samples were analyzed in triplicate. The procedure was conducted as quickly as possible to avoid dye degradation by exposure to light and alcohol evaporation. Trays with cuvettes were covered with aluminum foil until reading.

An additional cuvette was filled only with pure alcohol, and another cuvette was filled only with the filtered pure dye for spectrophotometer calibration (FEMTO 800 XI, São Paulo, Brazil). The absorbance baseline was determined by inserting the cuvette containing only alcohol into the reader. Then, a complete scan (300 to 800 nm) of the pure dye was performed to define its absorbance peak and the respective wavelength. This procedure was conducted each experimental day to calibrate the measures according to minimal absorbance variations from one batch to another. Once the wavelength corresponding to the absorbance peak was defined, all samples were read in the spectrophotometer accordingly. Samples were covered by aluminum foil up to read in the spectrophotometer. The reading was taken in a single run, without any gaps, to minimize the effect of eventual alcohol evaporation.

### 4.5. Statistical Analysis

Statistical analysis was performed using Microsoft EXCEL and GraphPad Prism version 6.0 for Windows. Normality was assessed by the Shapiro-Wilk test and by inspection of quartile-quartile plots (Q-Q plots). Levene’s test evaluated the homogeneity of variances, and Welch’s correction was applied to one-way ANOVA in cases of non-homogeneity. Outliers were evaluated by Q-Q plot inspection and removed if necessary. The individual values are expressed in the graphics. Tukey’s post-test was used to compare one group to another. The frequency of nauplii swimming in the bottom/surface was analyzed by *X*^2^. Two-way analysis of variance (ANOVA) with mixed models was used for a combined evaluation of the effects of different treatments and levels of GBH exposition. The total number of nauplii is indicated in the respective legends. *p* < 0.05 were considered significant. Figures were made using Prism 8.3 software.

## 5. Conclusions

Gly 6 cH treatment appears to produce protective effects on the development of *Artemia salina* exposed to GBH at low concentrations. These effects can be summarized as (a) an increase in nauplii vitality, (b) the selection of healthier nauplii during hatching, (c) an optimal protective effect in a specific range of water salinity, (d) the protective effect of Gly 6 cH can be seen as an increase in bio-resilience process (e) Coumarin 7 was identified as a potential physicochemical marker for Gly 6 cH activity in real-life conditions, (f) the theoretical concept of quantum coherence domains gives a plausible explanation to these findings.

## Figures and Tables

**Figure 1 ijms-24-09478-f001:**
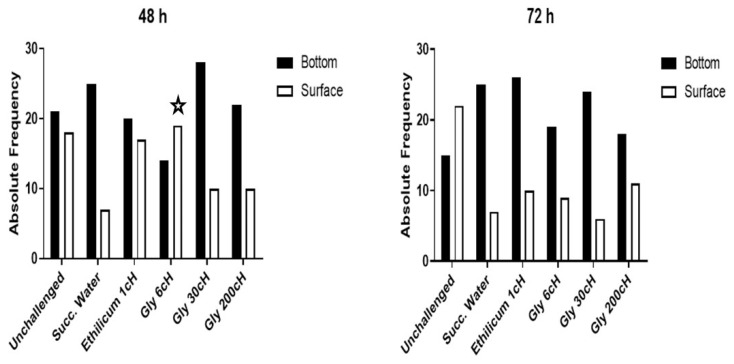
Absolute frequency of active nauplii in the deepest layers of the tube (Bottom) and the superficial one (Surface). At 48 h: *X*^2^, *p* = 0.01 between Gly 6 cH and succussed water (Succ. Water), indicated by the star (N = 403).

**Figure 2 ijms-24-09478-f002:**
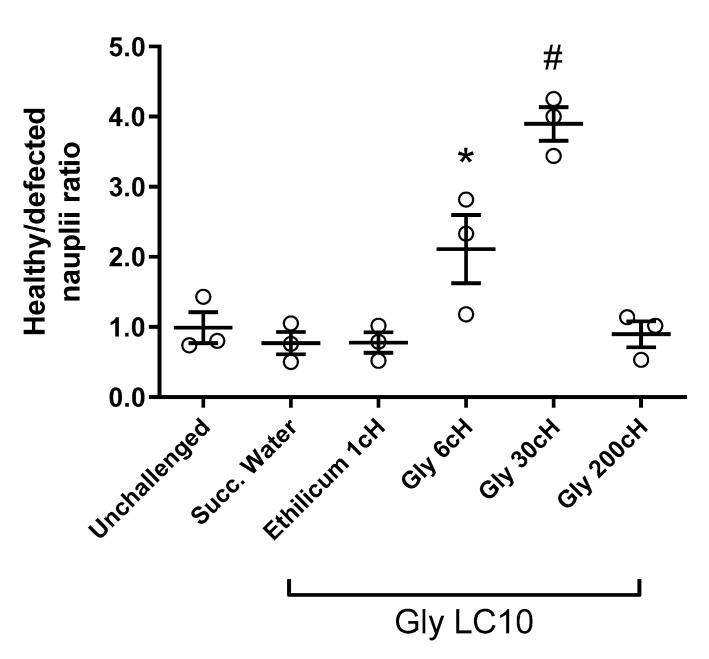
Healthy/defective ratio related to the number of nauplii per microscopic field after different treatments. Samples were made in triplicate. A significant treatment effect was observed: One-way ANOVA, F _(5, 12)_ = 22.02, *p* < 0.0001. Tukey post-test revealed that * *p* = 0.036 in relation and ^#^ *p* < 0.0001 in relation to the controls (cysts unchallenged and untreated, cysts treated with succussed water, and cysts treated with *Ethilicum* 1 cH). Data expressed as individual values, mean ± standard error (N = 1334).

**Figure 3 ijms-24-09478-f003:**
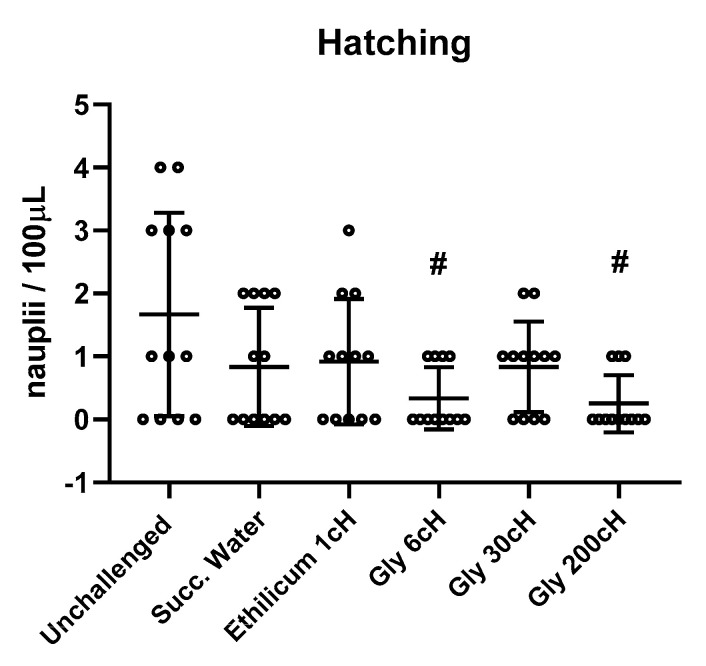
The population of living nauplii after cyst hatching (48 h). The population density was measured in numbers of nauplii per 100 microliters. One-way ANOVA, F _(5, 66)_ = 3.407, *p* = 0.0085. Tukey, ^#^
*p* ≤ 0.02 in relation to the unchallenged group. Values represent individual values, mean ± standard deviation (N = 59 nauplii distributed in 12 samples/datapoints of 100 µL per treatment).

**Figure 4 ijms-24-09478-f004:**
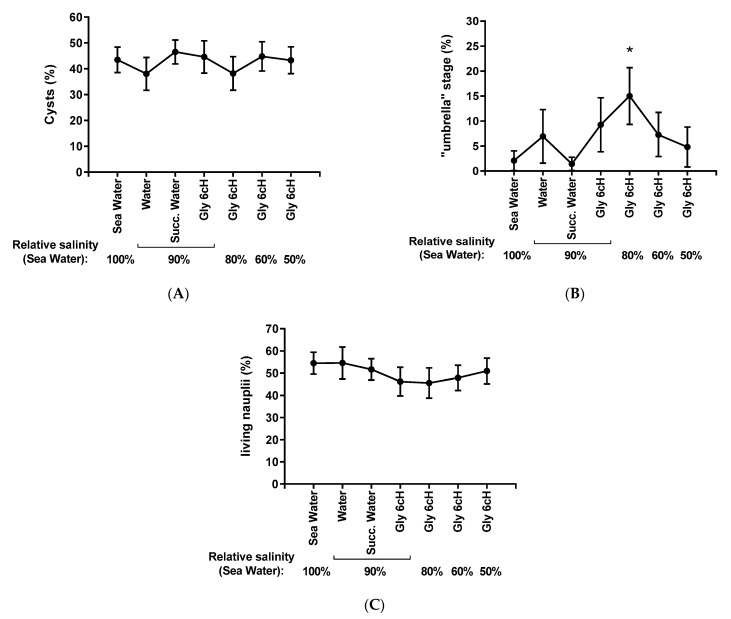
Percentage of cysts (**A**), “umbrella” stage (**B**), and living active nauplii (**C**) after 48 h of hatching process under different experimental conditions: exposed to 100% seawater; 90% seawater treated with 10% pure water, succussed water or Gly 6 cH; 80%, 60%, and 50% seawater treated with 10% Gly 6 cH. Two-way ANOVA with mixed models/Tukey made the statistical analysis, being: (**A**) F (_6, 298_) = 1.57, *p* = 0.155; (**B**) F (_6, 298_) = 4.61, * *p* < 0.001 in relation to other experimental conditions; (**C**) F _(6, 298)_ = 1.88, *p* = 0.083. Data expressed by mean ± standard error. N = 48,749.

**Figure 5 ijms-24-09478-f005:**
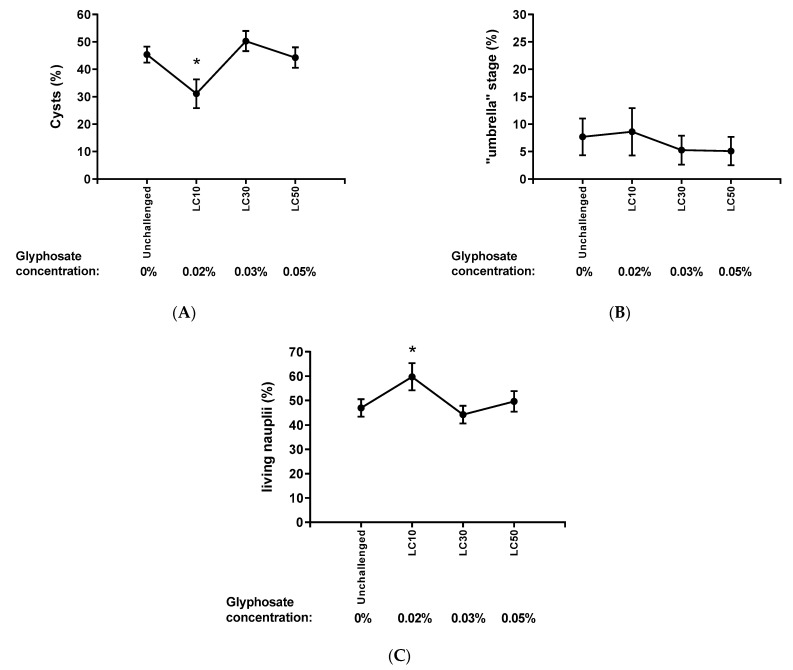
Percentage of unhatched cysts (**A**), “umbrella” stage (**B**), and living active nauplii (**C**) after 48 h of the hatching process under different exposition levels of Glyphosate, from zero to LC50 (0.05%). Two-way ANOVA with mixed models/Tukey made the statistical analysis. A complementary behavior was seen between (**A**) cysts percentage [F (_3, 298_) = 16.86, *p* < 0.0001. Tukey, * *p* < 0.05 compared to the other glyphosate concentrations] and (**C**) living nauplii percentage [F (_3, 298_) = 10.32, *p* < 0.0001. Tukey, * *p* < 0.05 compared to the other glyphosate concentrations], showing that LC10 was the ideal concentration to promote cyst hatching. No significant difference was seen regarding the “umbrella” stage (**B**) [F (_3, 298_) = 1.15, *p* = 0.329]. Data expressed by mean ± standard error. N = 48,749.

**Figure 6 ijms-24-09478-f006:**
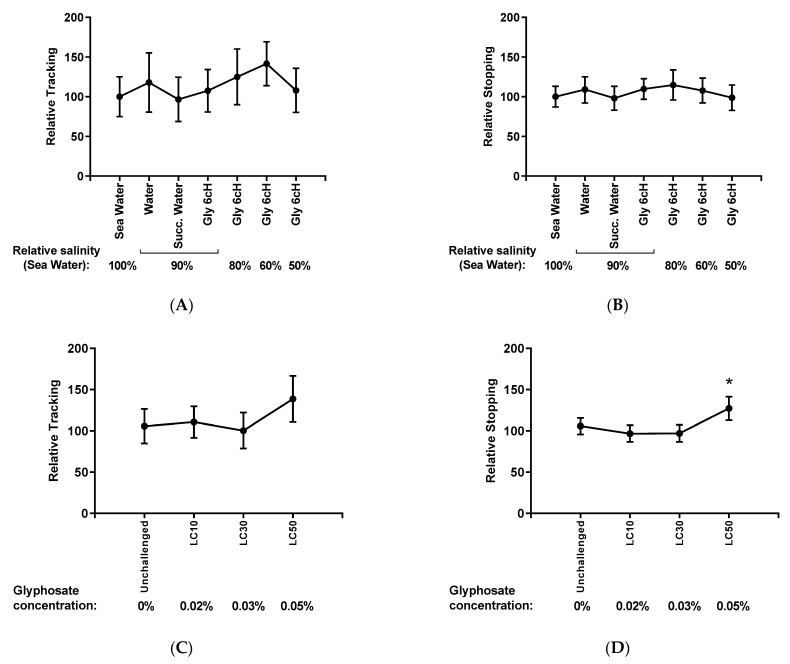
Frequency of locomotion (**A**) and (**C**) and stopping (**B**) and (**D**) of nauplii (in pixels) compared to the level of water salinity and Gly 6 cH treatment (**A**) and (**B**) and compared to the level of glyphosate exposition (**C**) and (**D**). The average obtained from unchallenged was considered as 100, and variations observed in the different treatments oscillated proportionally more or less. Two-way ANOVA with mixed models/Tukey made the statistical analysis. (**A**) F _(6, 210)_ = 1.470, *p* = 0.190; (**B**) F _(6, 210)_ = 1.303, *p* = 0.257; (**C**) F _(3, 210)_ = 1.290, *p* = 0.247; (**D**) F _(3, 210)_ = 5.886, * *p* = 0.0007 in relation to the other groups. Data expressed by mean ± standard error. N = 48,749.

**Figure 7 ijms-24-09478-f007:**
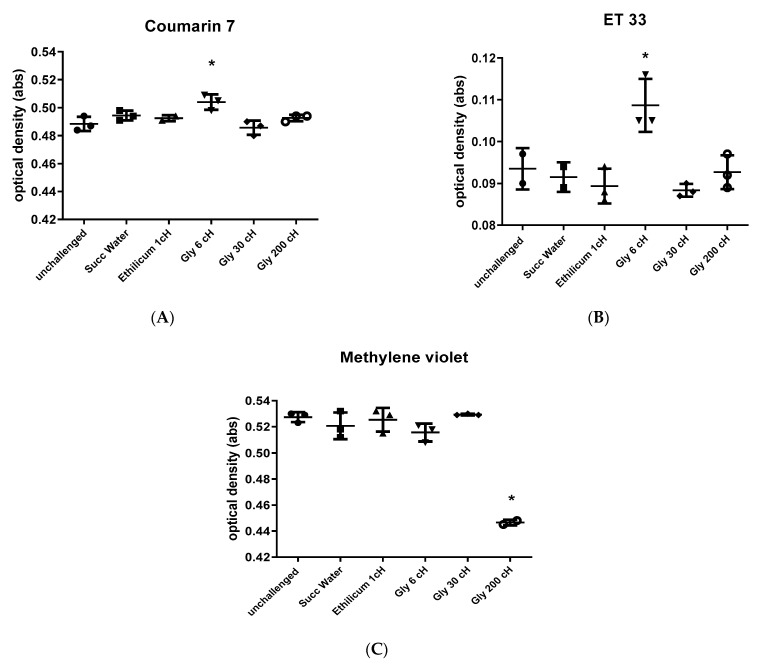
Absorbance (optical density, in nm) of Coumarin 7 (**A**), ET 33 (**B**), and Methylene violet (**C**) after interaction with samples of seawater treated with different GBH potencies, *Ethilicum* 1 cH, and vehicle. The baseline is shown as unchallenged. One-way ANOVA/Tukey, (**A**) F _(5, 11)_ = 6.327, * *p* = 0.0005 in relation to unchallenged and Gly 30 cH; (**B**) F _(5, 10)_ = 8.697, * *p* < 0.05 in relation to the other treatments; (**C**) F _(5, 11)_ = 47.66, * *p* = 0.001 in relation to the other treatments. Data expressed by individual values, mean ± standard deviation.

**Figure 8 ijms-24-09478-f008:**
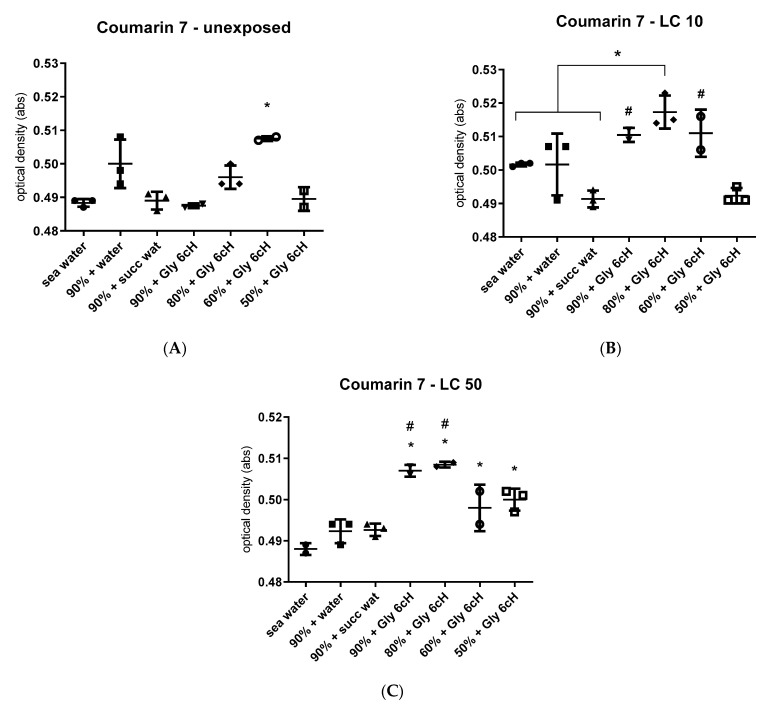
Absorbance (optical density, in nm) of Coumarin 7 after interaction with samples of seawater treated with Gly 6 cH in different seawater concentrations. (**A**) unexposed seawater; (**B**) seawater exposed to 0.02% glyphosate (LC10); (**C**) seawater exposed to 0.05% glyphosate (LC50). Water and succussed water were used as controls. The baseline is shown as unchallenged. One-way ANOVA/Tukey, (**A**) F _(6, 10)_ = 17.80, *p* < 0.000. Tukey, * *p* < 0.05 in relation to the other groups; (**B**) F _(6, 12)_ = 10.86, *p* = 0.003. Tukey, * *p* < 0.03; ^#^ *p* < 0.03 in relation to 90% seawater + succussed water; (**C**) F _(6, 10)_ = 17.92, *p* < 0.0001. Tukey, * *p* < 0.05 compared to seawater; ^#^
*p* < 0.002 compared to all controls (seawater, 90% seawater + water, 90% seawater + succussed water).

**Figure 9 ijms-24-09478-f009:**
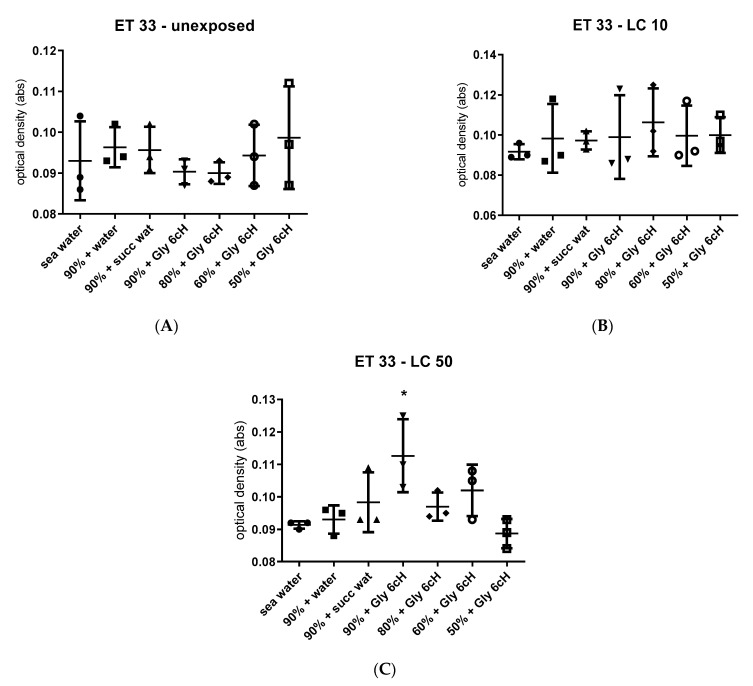
Absorbance (optical density, in nm) of ET 33 after interaction with samples of seawater treated with Gly 6 cH in different seawater concentrations. (**A**) seawater from unexposed cysts; (**B**) seawater from cysts exposed to 0.02% glyphosate (LC10); (**C**) seawater from cysts exposed to 0.05% glyphosate (LC50). Water and succussed water were used as controls. The baseline is shown as unchallenged. One-way ANOVA/Tukey, (**A**) F _(6, 14)_ = 0.556, *p* = 7575; (**B**) F _(6, 14)_ = 0.290, *p* = 0.9317; (**C**) F _(6, 14)_ = 4.056, *p* = 0.014. Tukey, * *p* < 0.05 compared to seawater.

**Figure 10 ijms-24-09478-f010:**
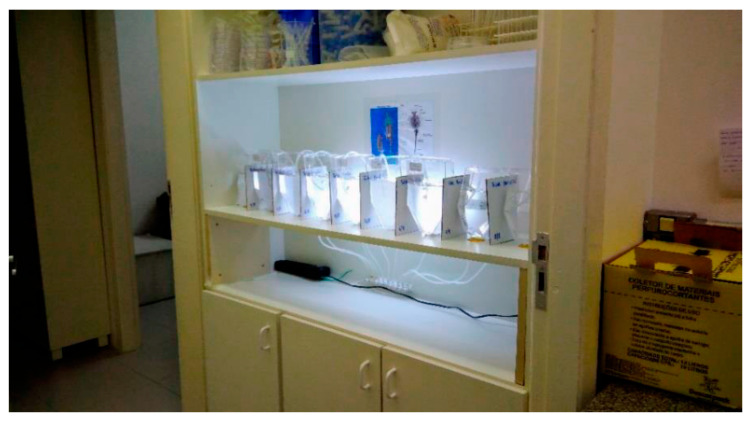
Culture bottles containing *Artemia salina* cysts in 400 mL of seawater enriched with *Saccharomyces cerevisiae* for nauplii feeding. A constant flux of oxygen was provided using a continuous air flow pump, and a constant luminosity was supplied for 48 h. A plate of mu-metal was used to separate one bottle from another, to avoid eventual field interference among them.

**Figure 11 ijms-24-09478-f011:**
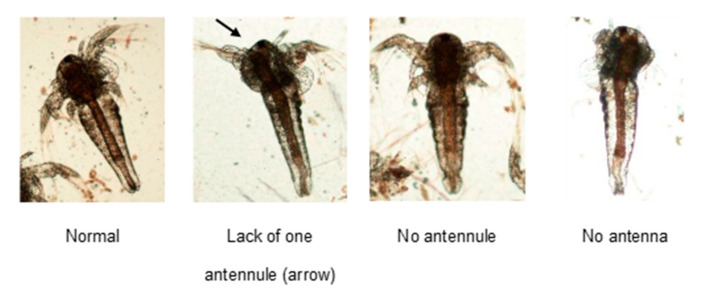
Representation of nauplii defeats found in the analyzed population distributed in 3 slides per group. Digital photomicrography obtained from an optical photomicroscope at 4× magnification. Arrow indicating the lack of an antennule.

**Table 1 ijms-24-09478-t001:** Experimental design of the plate-made assays.

Plate		Plate 1 (Glyphosate CL 10)	Plate 2 (Glyphosate CL30)	Plate 3 (Glyphosate CL50)	Plate 4 (Unchallenged Cysts)
Salinity					
180 µL of 90% seawater	20 µL Gly 6 cH	0.02%	0.03%	0.05%	zero
180 µL of 80% seawater	20 µL Gly 6 cH	0.02%	0.03%	0.05%	zero
180 µL of 60% seawater	20 µL Gly 6 cH	0.02%	0.03%	0.05%	zero
180 µL of 50% seawater	20 µL Gly 6 cH	0.02%	0.03%	0.05%	zero
180 µL of 100% seawater	20 µL pure water	0.02%	0.03%	0.05%	zero
180 µL of 100% seawater	20 µL succussed pure water	0.02%	0.03%	0.05%	zero
200 µL of 100% seawater (baseline)	-	0.02%	0.03%	0.05%	zero

**Table 2 ijms-24-09478-t002:** Solvatochromic dyes, the respective concentration (µM) in ethanol P.A., and the standard wavelength (nm) used in this spectrophotometric study.

Dye	CAS	Color in Alcohol	Molarity	Wavelength
REICHARDT’S DYE (ET30)	10081-39-7	violet	200 µM	550 nm
COUMARIN 7	27425-55-4	fluorescent green/yellow	25 µM	433 nm
RHODANINE [5—(4–DIMETYLAMINEBENZILIDENE)]	536-17-4	yellow	50 µM	457 nm
ET 33 (-) 2,6-DICHLOR-4-(TRIPHENILPIRIDINIUM-1-IL)-PHENOLATE	121792-58-3	orange	245 µM	484 nm
NILE RED	7385-67-3	pink	20 µM	550 nm
METHYLENE VIOLET	2516-05-4	purple	50 µM	600 nm
N, NDIMETHYLINDOANILINE	2150-58-5	blue	25 µM	595 nm

## Data Availability

The data presented in this study are available on request from the corresponding author since no public repository was used during the study execution. The individual values (X^2^ and one-way ANOVA) and the total number of nauplii (two-way ANOVA with mixed models) are indicated in the figures.

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
