# Peer review of "Highly Diluted Glyphosate Mitigates Its Effects on Artemia salina: Physicochemical Implications"

_ijms, 2023, doi:10.3390/ijms24119478_

Round 1

Reviewer 1 Report

The manuscript is of significance however there are the following queries that needed to be addressed:

What is the significance homeopathy in the present study as the term is used as a keyword?

The introduction should include a paragraph on herbicides and their application to pest control.

Figure 2: The change in morphology of nauplii can be due to the degradation after euthanasia over time. How the authors ruled out the change was exclusively due to the treatment?

It would be better if the 3.3.2. Experiment 2 is expressed in table format.

How was the instar I to instar V of nauplii larva identified?

The data for Solvatochromic dyes shown in Table 1. is not shown in the results. Does the study use all dyes in Table 1 or just the three dyes mentioned in the results?

The text needs to be revised for grammar check and all scientific names should be in italics.

Reviewer 2 Report

The manuscript is an experimental result of a lot of work with good and correct statistics; there are no serious comments to it.

Nevertheless, it can be noted that writing concentrations of 10-61 and 10-401 cause a smile only, since they do not make sense by definition. It is quite enough to specify one real concentration for Gly 6CH.

If the 2/5th parts of the column at the bottom of the tube belong to the bottom, why the 3/5th parts of the column are closest to water surface (instead of 4/5) - lines194-195.

The 200% scale in Fig. 8 for tracking and stopping is not clear.

The discussion of a single mechanism to explain the observed effects is not a scientific discussion, but rather a belief in also unfounded hypotheses. But this is at the level of understanding of the authors.

The manuscript can be published, preferably correcting minor comments without secondary reviewing.

Round 2

Reviewer 1 Report

The response from the authors towards the comments were satisfactory.

Minor language correction is still needed.

Author Response

A new (and last) English grammar and style review was done.